# Distress, Depression, Anxiety, and Concerns and Behaviors Related to COVID-19 during the First Two Months of the Pandemic: A Longitudinal Study in Adult MEXICANS

**DOI:** 10.3390/bs11050076

**Published:** 2021-05-13

**Authors:** Aldebarán Toledo-Fernández, Diana Betancourt-Ocampo, Alejandro González-González

**Affiliations:** Facultad de Psicología, Campus Norte, Universidad Anáhuac México, Universidad Anáhuac Avenue #46, Lomas Anáhuac, 52786 Huixquilucan, Mexico; aldebaran.toledofe@anahuac.mx (A.T.-F.); diana.betancourt@anahuac.mx (D.B.-O.)

**Keywords:** COVID-19, distress, depression, anxiety, Mexico, longitudinal study, two-wave design

## Abstract

We examined longitudinal differences in the severity of distress, depression, anxiety, and concerns and behaviors related to COVID-19 during the first two months of this pandemic, correlations between these variables, and interactions of distress with significant sociodemographics across waves. A longitudinal online survey was conducted in the State of Mexico, from 8 April to 27 May, 2020, in a sample of men and women between 18 and 60 years old, using: Impact of Event Scale-6, Patient Health Questionnaire-9, General Anxiety Disoder-7, and a questionnaire of concerns and behaviors related to COVID-19. Six hundred seventy participants were analyzed. Only a mild difference in distress was observed between the two waves and mild correlations of this variable with contagion in oneself and in a relative. Having a high-risk medical condition proved a considerable effect on distress within both waves. Perception of usefulness of preventive measures, concerns of contagion in a relative, and financial and security situations scored high within our questionnaire but did not change in the follow-up. We hypothesize that habituation to distressful events in the Mexican population (emergent resilience) might explain the absence of meaningful differences. Our research adds to the monitoring of mental health in Mexicans during the COVID-19 pandemic; its findings can serve to perform comparisons in other studies and for further meta-analyses.

## 1. Introduction

The coronavirus disease 2019 (COVID-19) has had a substantial impact on the mental health of the worldwide population, as was expected from the beginning in light of the evidence from other disease outbreaks of far lesser duration and scope [1,2,3]. Across the globe, research has consistently reported a worsening of mental health [4], particularly on psychological variables sensitive to highly unusual and routine-altering phenomena, such as a pandemic. In this study, we focused on three main variables: distress, depression, and anxiety.

Psychological distress is understood as a persistent feeling of overwhelm and vulnerability in relation to a potentially traumatic event, such as those that disrupts social functioning, like wars, natural disasters, and epidemics. In this context, distress is a variable that is commonly assessed as posttraumatic stress symptomatology, characterized by intrusive thoughts, avoidance of the distressful event, and hyperarousal mainly manifested in cognitive alterations, such as dysprosexia and hypervigilance [5]. Depression is characterized by a persistent feeling of sadness, apathy, and anhedonia as the core symptomatology, with other secondary problems, such as sleep and eating disturbances, psychomotor retardation, dysprosexia, low self-esteem, and suicidal ideation [6]. Anxiety is understood as persistent worry and fear about everyday situations, often expressing in restlessness, irritability, and persistent catastrophic ideation [7]. Both depression and anxiety can manifest themselves exclusively, comorbidly [8], or being part of the symptomatology of psychological distress, particularly in the context of stressful events [5,9,10].

Assessing distress, depression, and anxiety is a crucial task since, when sufficiently intense, they can lead to impairments in everyday-life functioning, such as decreased work or academic performance, relationship strain, family dysfunction, higher risk of substance use, or even reduction in overall health [8,11]. When considering that a global common stressful phenomenon could increase the manifestation of these mental health problems not at the individual level but at a population level, the importance of monitoring them is highlighted even further. Because of this, independent of each other or in conjunction, distress, depression, and anxiety have been notoriously monitored in several populations worldwide as the pandemic evolves, converging on the main finding that they have tended to score high or increased at different points during the pandemic and that several factors, such as female gender, age group (adolescents, young adults, and elderlies), higher education [12,13], lower-income or socioeconomic status [14], substance use [15], preexisting medical and mental health conditions [16], experiences around the disease [17], knowledge about it [13,18], and even contact with nature [19], among many others [20], may play an interacting role.

As the pandemic evolves, the evidence is accumulating rapidly, and researchers from practically all countries are making efforts to keep actualized the dynamics of the associated mental health variables. In the case of Mexico, for which our study stands as one of those efforts, already published research with samples from different regions of the country has found similar worsening of mental health. A study found that assessing COVID-19 as a high-risk disease and perceiving oneself as being at risk of contracting it had only small-to-moderate effects on the perception of the consequences of the disease, the emotional impact, and the adherence to preventive behaviors [21]. Another study found no differences in worry and perceived risk relative to educational attainment, occupational demandingness, or isolation [22]. A study from our own research group [23] conducted on older adults found that depressive symptoms were more frequent among women 80 years or older compared to men of the same age or both genders aged 60 to 79.

As informative as these studies are, they had a cross-sectional design, bypassing the analysis of possible changes in the studied psychological variables that most likely have occurred as the pandemic evolves. As suggested early in the literature concerning this pandemic [3,24], it is of crucial importance to keep monitoring the mental health of populations in order to allocate health care resources. Because of this, we conducted a follow-up study during the first two months of the COVID-19 pandemic.

In Mexico, these first two months could be characterized by the following important events: 24 March, the Mexican Ministry of Health published the preventive measures for COVID-19 [25]; 30 March, the General Health Council declared a health emergency due to COVID-19 [26]; 31 March, extraordinary actions were published to address this emergency, including the immediate suspension of non-essential activities in the public, private, and social sectors, aimed at reducing the collapse of the health system [27]. These measures were said to last until 30 April [27]. At the beginning of this study (8 April), Mexico had 2785 cases and 141 deaths from COVID-19 [28]; at the time of the follow-up (11 May) there were already 36,327 confirmed cases and 3573 COVID-19 deaths [29]. By this time, arguably, these and other associated events may have had some psychological impact on the Mexican population.

Longitudinal mental health changes during a one-to-four month period have already been reported for different countries [30,31,32,33,34], recognizing the importance of keeping track of the psychological implications of this global event. Aiming to provide local information concerning psychological distress associated with the COVID-19 pandemic, the objectives of the current report were to examine: (1) longitudinal changes in the severities of distress, depression, anxiety, and concerns and behaviors related to COVID-19 during the first two months of this pandemic in Mexico, (2) correlations between these variables, and (3) moderating effects of sociodemographic characteristics on distress across waves, in an adult sample from a metropolitan area in the central region of Mexico.

As the general literature on social stressful events describes, and more specifically the growing body of evidence accumulated during the pandemic, we hypothesize increases in the severities of distress, depression, and anxiety, as well as in concerns and behaviors related to COVID-19, from baseline to follow-up. We also expect to find significant moderate correlations between these variables, as examples: positive correlations between distress and concerns about contagion in oneself and contagion in a relative; positive correlations between distress and financial and security worries. Our last general hypothesis is that sociodemographics will serve as moderating variables between baseline and follow-up. Specifically, we expect increases in distress between baseline and follow-up in three groups: women, older groups of age, and individuals with children. Concerning the former, literature on personality has clearly demonstrated a greater inclination in them towards negative emotionality [35], and within the context of the COVID-19 pandemic, there is evidence of a greater tendency in women to experience feelings of loneliness closely related to family and care responsibilities [36]. Regarding the differences found by age groups, a recent study [37] reported that older adulthood was associated with the perception of greater risks of dying if COVID-19 was contracted, but with the perception of a lower risk of contracting COVID- 19, being quarantined, or running out of money, as well as lower levels of depression and anxiety and negative financial worries; this is interesting when considering the more optimistic view in older adults at the beginning of the pandemic and a better response to daily stressors [38,39]. Regarding the last hypothesis, another recent study [40] found that the concern about knowing oneself, a relative, or a significant other to be infected with COVID-19 is an issue that emerges as one of the great concerns experienced with this phenomenon. In addition, some authors suggest that people’s concerns about the probability of getting sick from COVID-19 can influence their attitudes and behaviors [41]. These concerns are due to the rapid increase in the spread of the virus, and it is understandable that people do not want to be infected or that their loved ones become infected due to the important health consequences that could lead to being infected or even dying [42]. In this regard, a study [43] found significant correlations between concern about the contagion of COVID-19 with anxiety, well-being, and perception of health in general in a population of young people and adults.

This study has the ultimate purpose of contributing to the efforts to keep monitoring the psychological implications of the COVID-19 pandemic within a local scope since (at the time this was written) there are no published longitudinal studies within the Mexican population and within the global framework, since its findings can add to the general evidence in further systematic reviews. Considering how fast this pandemic keeps evolving, analyses of multiple windows of time across different populations are of great need for knowledge on this topic.

## 2. Methods

### 2.1. Study Design and Participants

This study is a two-wave longitudinal online survey with a non-probabilistic sample, recruited by convenience sampling technique. Participants were men and women between the ages of 18 and 60 years old, living in the State of Mexico (a large territory in the Center of the country, comprising 125 municipalities, some of them being the most densely populated in the country), who were contacted by the staff of the State Council for Women and Social Welfare (Consejo Estatal de la Mujer y Bienestar Social (CEMyBS)), via email or telephone who were registered in the database of their program’s beneficiaries.

The first data collection lasted 10 days, beginning on 8 April 2020, coinciding with the end of week two of the national lockdown. The second data collection began on May 11th and finished on 27 May 2020.

### 2.2. Measures

#### 2.2.1. Sociodemographics

The questionnaire to collect this information included self-reported items about gender (male or female), age group (18–25, 26–39, 40 or more), educational attainment (no degree to middle school, high school, technical degree/college, and specialty/masters/doctorate), relationship status (married, single, free union, divorced/separated/widowed), employment (not or currently employed), having children (none, underaged children, and adult children), and current high-risk medical condition (none or any of the following diagnoses: hypertension, diabetes, cancer, respiratory diseases, autoimmune disease or immunosuppression, obesity, or dyslipidemia).

#### 2.2.2. Distress

This variable was measured with the Impact of Event Scale-6 (IES-6), which is a brief form of the widely used Impact of Event Scale-Revised [5,44]. It assesses self-report of posttraumatic distress reactions to specific events. The IES-6 includes two items for each of the dimensions of posttraumatic stress: intrusion, avoidance, and hyperarousal, and five response options ranging from 0 (“Not at all”) to 4 (“Extremely”). A total score is computed, with higher values indicating more severe distress and values between 10 to 13 points as possible cutoffs for detection of significant distress (or posttraumatic distress disorder) from which we used the 13-points score because of its specificity of 0.99 (sensitivity = 0.55; overall efficiency = 0.87) [5]. To provide a more fine-grained analysis, we also computed the three subscale scores characterizing symptoms of significant event-related psychological distress: intrusion, avoidance, and hyperarousal, each comprising a pair of items from the total six, and scores from 0 to 4 total points [5].

For this study, we employed corresponding Spanish-translated items [45], and we instructed the respondents to answer them considering the COVID-19 pandemic as the potentially distressful event, as follows: “Some people often experience difficulties during distressful events. In the following statements, think about the last 7 days and how distressful the situation we are living in due to the coronavirus pandemic has been for you.” Some examples of the statements are: “I thought about it when I didn’t mean to”, “I felt watchful or on-guard.” [5]. Testing the reliability of the total score of this instrument within our samples, we found Cronbach’s α = 0.86 for wave 1 and α = 0.89 for wave 2, suggesting good internal consistency.

#### 2.2.3. Depression

For this variable, the Patient Health Questionnaire (PHQ-9) was used. This self-administered 9-items scale inquires about specific depressive symptoms corresponding to DSM-IV criteria in the last two weeks, according to a Likert scale with 0 to 3 values (e.g., “Over the last 2 weeks, how often have you been bothered by the following problems? […] Feeling down, depressed. […] Moving or speaking too slow or foo fast.”). A total score is then computed, ranging from 0 to 27, with upper values indicating a higher severity of depression. For this study, a categorical diagnosis was determined considering a value of 15 or more for moderate to severe depression according to the original cutoff scores [6].

The PHQ-9 is a very common scale in clinical research worldwide; evidence of its validity has been reported for Latin American and Mexican populations [46,47], and it has been used in online surveys [48,49]. Satisfactory internal consistency for this instrument was observed within wave 1 (Cronbach’s α = 0.89) and wave 2 (Cronbach’s α = 0.91).

#### 2.2.4. Anxiety Severity

It was measured with the Generalized Anxiety Disorder-7 (GAD-7), which is a seven-item self-administered scale that was developed to assess general anxiety disorder following the DSM-IV symptom criteria [50] and can be used to evaluate other forms of anxiety [7] (e.g., “Over the last 2 weeks, how often have you been bothered by the following problems? […] Feeling nervous, anxious or on edge. […] Being so restless that it is hard to sit still.”). The GAD-7 has 0 to 3 response options, with a total score between 0 and 21. Higher scores indicate more severe anxiety. As with the PHQ-9, a categorical diagnosis was determined considering a value of 10 or more for moderate to severe anxiety according to the original cutoff scores [50].

Evidence of good internal consistency and validity has been reported for Mexican samples [51,52], and the scale has been used in online surveys [53]. As with the other instruments, good internal consistency was obtained for both waves (Cronbach’s α = 0.91, and α = 0.93, respectively).

#### 2.2.5. Concerns and Behaviors Related to COVID-19

These were evaluated with a 12-item questionnaire that was designed specifically for this study with explorative purposes since no standard measure for the assessment of concerns and behaviors related to COVID-19 was available at the time this study was designed. The questions were elaborated by the authors of the current study (two of them specialized in social psychology, and one specialized in cognitive psychology), aiming to assess concerns and behaviors in five categories: monitoring and usefulness of prevention measures against COVID-19; information on the pandemic; concerns about COVID-19; impact of COVID-19; and family care (Table 1). These categories were considered partially following one of the first published studies on the subject matter in a Chinese sample [54].

The design and development of the questionnaire consisted of the following phases, all executed by consensus of the three authors: (1) an original pool of 20 items was written; (2) considering time constraints of the online survey and aimed at reducing the friction caused to participants in responding, the total number of items was reduced to only twelve; (3) sentence grammar was refined to better convey the meaning of each item; (4) further grammar refinement was performed after piloting the questionnaire with a pair of colleagues. Response options were set at a 0 to 10 scale, with higher values intended to reflect higher frequency or intensity of the assessed concern or behavior. The granulation of the scale was established in this fashion to allow for a more continuous variability of the responses. Each item was considered independently in the main analyses (as was also the procedure in the study by Wang et al. [54]).

This questionnaire was recently used in a study with older adults [23], proving significant associations with psychological distress and interaction with relevant sociodemographic characteristics, such as gender and group age. The questionnaire is not intended as an instrument to be used as a standard measure but rather was designed specifically to evaluate these concerns and behaviors in the context of our study, considering limitations of time in the context of the quick development of the pandemic. In the current study, analysis of internal consistency between the total items of the questionnaire showed Cronbach’s α = 0.83 for the first wave and α = 0.86 for the second wave, which suggests acceptable consistency.

### 2.3. Procedure

Individuals were contacted by the staff of the CEMyBS through email and telephone, inviting them to participate in the online survey that was sent to them via email. The survey was designed and conducted using Google Forms and took 10–15 min to complete. Researchers had no access to the database of the CEMyBS, so they were blinded to personal information that could compromise confidentiality. As well, data from the measures employed in this study were not shared with the CEMyBS. At the end of the survey, each participant was requested to voluntarily provide his/her email address, or the matching of baseline and follow-up measures, emphasizing that it would strictly be used for the follow-up measurement.

### 2.4. Statistical Analysis

Descriptive statistics were computed for sociodemographic characteristics, total scores of the IES-6, the PHQ-9, the GAD-7, and each of the items exploring concerns and behaviors related to COVID-19. A Paired-samples *t*-test was used for comparing the numerical results of these instruments, and Pearson’s product-moment correlations were utilized to test for associations between these variables. An independent-samples *t*-test and one-way ANOVA were used to test for differences in distress within sociodemographics for each wave. ANOVA for repeated measures was used to analyze interactions between distress and sociodemographic characteristics that proved meaningful effect size in the paired-samples *t*-test.

Missing values were handled for each inferential analysis by excluding cases pairwise. All the inferential results were judged significant at a two-tailed *p* < 0.05, and effect sizes were considered of medium magnitude for discussion when r ≥ 0.25, d ≥ 0.50, or η^2^ ≥ 0.06 [55]. Statistical analyses were performed using JASP 0.14.1.0 [56].

### 2.5. Ethical Considerations

The survey included a brief informed consent describing the general objective of the study, the subject matter addressed by the questions, voluntariness, and confidentiality of participation, and the institutions involved in the implementation of the study. Before starting the survey, individuals were asked to check the box for consent.

The research protocol was approved by the Bioethics Committee of the Faculty of Health Sciences—[BLINDED] (202003, CONBIOETICA-15-CEI-004-20160729).

## 3. Results

### 3.1. Flow of Recruitment and Sociodemographic Characteristics

A total of 670 individuals were analyzed after excluding 118 cases (14.97%) due to inconsistent data between the two waves (e.g., mismatched reported emails). Within the final sample, 390 (58.20%) reported being female and 280 (41.49%) being male, with ages distributing as follows: 18–25 (16.41%), 26–39 (44.92%), 40 or more (38.65%). Forty percent of the participants reported being married, 35.82% being single, 15.07% in free union, and 9.09% described themselves as divorced/separated/widowed. Almost half of the participants (47.76%) reported having underaged children, and 20.44% adult children. Concerning educational attainment, distributions were: 11.94% no degree to middle school, 19.40% high school, 60.44% technical degree/college, and 8.2% specialty/masters/doctorate. With regards to occupation, a minority of wave 1 (8.35%) and wave 2 (9.10) reported being unemployed. Lastly, 26.56% reported having at least one high-risk medical condition.

### 3.2. Within- and between-Waves Difference Analyses

Concerning psychological variables, the following percentages of participants met cutoff criteria for diagnosis in waves 1 and 2, respectively: 27.61% and 21.94% for significant distress; 5.22% and 6.26% for moderate to severe depression; and 11.94% and 12.23% for moderate to severe anxiety. Only a weak difference in distress (d = 0.24) was observed between the two waves, with participants in the follow-up scoring 1.03 lower than in the baseline. When analyzing these differences by subscales of the IES-6, intrusion and avoidance displayed scores closer to the maximum of four points and, and all three components proved significant decrements from baseline to follow-up with low effect size, with intrusion proving the highest magnitude (d = 0.26) (Table 1).

Significant (*p* < 0.05) differences in distress within demographic characteristics were found in both waves, with higher scores relative to upper age groups, having children, and having a high-risk medical condition. Only within wave 2 did women show higher values of distress (Table 2). All these differences were of small magnitudes, except for having a high-risk medical condition, which proved close to medium effect size.

As shown in Table 1, in both waves, values for concerns and behaviors related to COVID-19 ranged from 1 to 10 points but tilted high for all the explored dimensions, particularly for perception of usefulness of preventive measures (items 2 and 4), for concerns of contagion in a relative (item 8), and regarding financial and security situations (items 9 and 10). Among all the items, compliance with hygiene and social distancing measures (items 1 and 3) had the lowest values in wave 1, increasing significantly in the follow-up (d = −0.37 for item 1, and d = −0.35 for item 3). No other difference with a non-negligible effect size was observed in this set of items.

### 3.3. Correlation Analyses

Most of the significant correlations between concerns and behaviors related to COVID-19, anxiety, depression, and distress were weak (r ≤ 0.22) in both waves, except between the latter and concerns regarding contagion in oneself (wave 1: r = 0.35, *p* < 0.001; wave 2: r = 0.31, *p* < 0.001) and in a relative (wave 1: r = 0.31, *p* < 0.00; wave 2: r = 0.27, *p* < 0.001).

### 3.4. Interaction Analyses

Considering the variables that proved between-waves differences in the paired-samples *t*-test as independent outcomes (IES-6, and items 1 and 3 of concerns and behaviors related to COVID-19), no significant interactions with gender, age group, having children, or having a high-risk medical condition were found (Table 3), meaning that no change was exercised by these variables from the first to the second wave. This lack of significant interactions remained when analyzing by IES-6 subscales.

## 4. Discussion

### 4.1. Longitudinal Changes in the Severities of Distress, Depression, Anxiety, and Concerns and Behaviors Related to COVID-19

The current study had the major goal of providing longitudinal information regarding distress, depression, and anxiety towards COVID-19 in adult individuals from a metropolitan area in the central region of Mexico, during the first two months of the pandemic, thus adding to the global efforts to keep monitoring psychological variables related to this event. Overall, we found that around a fourth of the sample, both in wave 1 and wave 2, met criteria for significant distress associated with the pandemic, with the total score of the IES-6 diminishing 1.07 points after two months, which implies a weak effect size (Table 4). When further analyzing distress by its components, we found that intrusion and avoidance showed the highest intensities and that all three components diminished mildly in the follow-up, suggesting persistence of overall symptomatology of distress. Intrusive distress thoughts seemed to have diminished the most, but there is not a clear-cut pattern as to further discuss this. Other studies using the IES-6 in relation to other potentially distressful events [57,58,59] have not reported differentiated patterns across scores of the subscales of this instrument, perhaps because of their low coverture (only two items per subscale).

Diagnoses of moderate to severe depression and anxiety were met by ~5.5% and ~12%, respectively, but with no significant changes between the two waves and only weak correlations with concerns and behaviors related to COVID-19. About these, we found high values in all concerns but between-waves differences only relative to adherence to hygiene measures and social distancing, with increments of low effect size. Longitudinal studies analyzing changes during a similar window of time (one to three months) have consistently reported increments in indicators of distress with some differences in the levels of severity, the specific components of distress, and the given explanations for the findings [15,32,34,60].

### 4.2. Associations between Distress, Depression, Anxiety, and Concerns and Behaviors Related to COVID-19

In our sample, the low correlation of depressive and anxiety symptoms with specific concerns about COVID-19 suggests that those symptoms were premorbid and possibly independent of the distressful event. This could find reaffirmation in the fact that depressive and anxiety symptoms did not change during the first two months of the pandemic. It is also possible, however, that the lack of changes could be due to insufficiency of these measures to register symptoms in association with the experience of the pandemic, as indeed the items on these scales (PHQ-9 and GAD-7) are written without any context other than timeframe (last two weeks).

In some contrast, the meaningful correlation between the IES-6 and the concerns and behaviors related to COVID-19 could indicate that, overall, Mexican individuals in our sample were indeed experiencing a moderate degree of distress, but it did not reach dysfunctional levels. Another study [21] on Mexican individuals converge with our results, for it reports that assessing COVID-19 as a high-risk disease and perceiving oneself as being at risk of contracting it had small to moderate effects on the perception of the consequences of the disease, the emotional impact, and the adherence to preventive behaviors. Furthermore, since the first wave, the prevalence of distress in our sample was considerably lower (27.61%) than in a Spanish (~49%) and Chinese (~31%) sample [30,46] and decreased (21.94%) after two months. As none of the sociodemographic characteristics that we collected explained this change, we hypothesize that possible habituation to distressful events in the Mexican population (due to regular exposure to financial insecurity and criminality) in the form of emergent resilience [61] might help to explain this comparatively low prevalence. This hypothesis could be reinforced considering that, by the time follow-up data was being collected, there were already 36,327 confirmed cases and 3573 COVID-19 deaths [29], and the lockdown and social distancing measures, as with almost all other Western countries, were not being withdrawn but rather projected to last for some months more, and all of these circumstances that could be expected to increase psychological distress. Further studies in the Mexican population or somewhat alike Latin American countries (such as Brazil, Chile, Argentina, or Colombia) are needed in this regard to test our hypothesis of emergent resilience.

### 4.3. Differences in Distress Related to Sociodemographics

When exploring differences between subgroups of sociodemographic variables, we found some commonalities with other studies, mainly higher distress in women, individuals aged 26 or more, individuals with children [13,54,62,63], and mostly in relation to having a medical condition of high risk for COVID-19, such as hypertension, diabetes, cancer, respiratory diseases, autoimmune disease or immunosuppression, obesity, or dyslipidemia, which has also been found in old-aged Mexican individuals [23], and samples from other countries, likely because of increased risk perception, which tends to be in part a function of how personal the experience related to COVID-19 is to the individual [64]. Except for having children, which we hypothesize as only being significant regarding underaged children, all the demographics showed the expected effects as declared in our hypotheses within each wave; however, they did not contribute to the decline of distress in the follow-up, according to our data, and express themselves with rather small magnitudes (d < 0.50, or η^2^ < 0.06). Concerning the lack of differentiation between having underaged or adult children in the ANOVA, we believe it could be explained because adult children is a very wide category that could include adult children still living at home, in which case it is possible that the hypothesized explanation (e.g., fear of leaving them without parental care or distress because of home-schooling) stands as plausible for this finding.

### 4.4. Limitations

First, the sample was composed of individuals registered in welfare programs, which could have biased the socioeconomic status within the middle- to low-income strata; these warn against generalization to individuals living in the central region of Mexico since it is characterized by marked socioeconomic differences. A second limitation was the loss of 14.97% of follow-up responses due to inconsistent matching data (e.g., email address), which could have decreased the robustness of our findings; however, we gained in minimizing the probability of spurious findings or inconsistent results by analyzing only perfectly matched cases. Finally, the sample size was obtained via non-probabilistic sampling, which restricts the generalizability of the findings; however, as the sample was composed of individuals from one of the most densely populated states in Mexico, and all being registered in a welfare program, it is likely that individuals were mostly representative of low-income stratum, the most prevalent socioeconomic in the country.

To our knowledge, this study is the first longitudinal examination of distress, depression, and anxiety in Mexican individuals during the first two months of the pandemic, and though this stands as a strength in our study, the window of time is also an inevitable limitation, for it captured only a small timeframe of a long and rapidly changing phenomena, and thus the observed psychological variables could easily have changed substantially by the time this was written. However, as we declared as justification of this study, providing empirical evidence of the psychological impact of this pandemic through different windows of time and across diverse populations is of critical importance.

### 4.5. Conclusions

Overall, our study adds to the monitoring of mental health in Mexican individuals during the initial stage of the pandemic in Mexico. The identified association between distress and some sociodemographic factors could guide the allocation of care resources (e.g., targeting primary mental health care to, for example, older women with medical conditions who also have children). Finally, findings here can serve to perform comparisons in other studies also interested in similar variables and for further meta-analyses on the psychological implications of COVID-19 in Mexican populations.

## Figures and Tables

**Table 1 behavsci-11-00076-t001:** Differences for concerns and behaviors related to COVID-19, distress, depression, and anxiety (N = 670).

	Wave 1	Wave 2	*t* (df), Cohen’s *d*
	Mean (SD)	Mean (SD)	
Monitoring and usefulness of prevention measures against COVID-19			
1. How often have you followed the recommended hygiene measures (e.g., constantly washing your hands, using alcohol for hands, disinfecting objects, using face masks, etc.) for the prevention of the coronavirus?	8.45 (1.51)	8.97 (1.28)	−9.57 (669) ***, −0.37
2. How useful do you consider these hygiene measures to be to avoid getting coronavirus?	9.08 (1.44)	9.18 (1.35)	−2.00 (669) *, −0.07
3. How much have you complied with the measures of social distancing?	8.05 (1.85)	8.67 (1.52)	−9.16 (668) ***, −0.35
4. How useful do you consider social distancing to be against the pandemic?	9.10 (1.44)	9.15 (1.42)	−1.14 (668), −0.04
Information on the pandemic			
5. How often do you search for information about the course of the pandemic?	8.23 (1.95)	8.09 (1.97)	1.76 (668), −0.008
6. How confident are you in the information you receive about the course of the pandemic?	7.54 (2.07)	7.55 (2.13)	−0.30 (668), −0.01
Concern about COVID-19			
7. How worried are you about getting the coronavirus?	8.66 (2.03)	8.71 (1.97)	−0.90 (668), −0.03
8. How concerned are you that a family member will get coronavirus?	9.31 (1.48)	9.22 (1.50)	1.38 (664), 0.05
9. How concerned are you about your financial situation as a result of the pandemic?	9.39 (1.37)	9.19 (1.58)	3.23 (668) ***, 0.12
10. How concerned are you about the security situation in your locality following the pandemic?	9.25 (1.29)	9.12 (1.40)	2.26 (668) *, 0.08
Impact of COVID-19			
11. How much has your daily life been affected by the pandemic?	8.62 (1.69)	8.73 (1.65)	−1.5 (668), −0.05
Family care			
12. How much has your family been aware of you since the contingency (phone calls, providing food, medicine, etc.)?	8.17 (2.27)	8.49 (1.96)	−3.95 (668) ***, −0.15
IES-6 total score	9.72 (4.54)	8.65 (4.86)	6.33 (669) ***, 0.24
Intrusion	3.49 (1.77)	3.02(1.84)	6.90 (669) ***, 0.26
Avoidance	3.23 (1.69)	2.93 (1.74)	4.44 (669) ***, 0.17
Hyperarousal	2.98 (1.62)	2.69 (1.74)	4.69 (669) ***, 0.18
PHQ-9 total score	4.87 (4.93)	4.86 (5.17)	0.08 (669), 0.003
GAD-7 total score	4.04 (4.36)	3.93 (4.45)	0.77 (669), 0.03

Notes: * *p* < 0.05; *** *p* < 0.001; Abbreviations: COVID-19 = coronavirus disease; GAD-7 = General Anxiety Disorder-7; IES-6 = Impact of Event Scale, abbreviated version; PHQ-9 = Patient Health Questionnaire-9.

**Table 2 behavsci-11-00076-t002:** Analyses of differences in distress (IES-6 total score) within sociodemographic characteristics for each wave ^a^.

	Wave 1	Statistical Difference	Wave 2	Statistical Difference
	Mean (SD)		Mean (SD)	
Gender				
Male	9.25 (4.70)	*t* (668) = −0.94, *d* = −0.07	8.16 (4.90)	*t* (668) = −2.22 *, *d* = −0.17
Female	9.86 (4.41)		9.00 (4.81)	
Age group				
18–25	8.28 (4.56)	*F* (2, 667) = 8.02 ***, *η*^2^ = 0.02 (post hoc: 18–25 < 26–39, 40 or more)	6.98 (4.50)	*F* (2, 667) = 8.02 ***, *η*^2^ = 0.02 (post hoc: 18–25 < 26–39, 40 or more)
26–39	9.87 (4.06)		8.89 (5.02)	
40 or more	10.15 (4.35)		9.08 (4.70)	
Having children				
Adult children	10.19 (4.04)	*F* (2, 667) = 5.12 **, *η*^2^ = 0.01 (post hoc: No children < Adult, Young children)	9.23 (4.44)	*F* (2, 667) = 5.70 **, *η*^2^ = 0.01 (post hoc: No children < Adult, Young children)
Underaged children	10.06 (4.61)		9.01 (4.93)	
No children	8.90 (4.64)		7.73 (4.92)	
Having a high-risk medical condition				
No diagnosis	9.19 (4.34)	*t* (668) = −5.07 ***, *d* = −0.44	8.02 (4.69)	*t* (668) = −5.65 ***, *d* = −0.49
With diagnosis	11.17 (4.76)		10.38 (4.93)	

Notes: * *p* < 0.05; ** *p* < 0.01; *** *p* < 0.001. ^a^ Only demographics that proved statistically significant difference in at least one wave are reported.

**Table 3 behavsci-11-00076-t003:** Correlation analyses event-related distress, depression, and anxiety, and concerns and behaviors related to COVID-19.

	IES-6				PHQ-9	GAD-7	Items of the Questionnaire of Concerns and Behaviors Related to COVID-19										
	Total	Int	Avo	Hyp			1	2	3	4	5	6	7	8	9	10	11
Wave 1																	
1	0.12 **	0.11 **	0.10 **	0.12 **	−0.15 ***	−0.10 **	--										
2	0.16 **	0.13 ***	0.12 ***	0.17 **	−0.04	−0.006	0.48 ***	--									
3	0.07	0.06	0.05	0.06	−0.08 **	−0.06	0.46 ***	0.42 ***	--								
4	0.20 ***	0.18 ***	0.15 ***	0.20 ***	−0.01	0.02	0.40 ***	0.66 ***	0.46 ***	--							
5	0.19 ***	0.22 ***	0.11 **	0.16 ***	−0.04	−0.03	0.47 ***	0.45 ***	0.34 ***	0.47 ***	--						
6	0.17 ***	0.20 ***	0.10 ***	0.15 ***	−0.12 **	−0.07 *	0.36 ***	0.47 ***	0.35 ***	0.45 ***	0.58 ***	--					
7	0.35 ***	0.37 ***	0.27 ***	0.29 ***	0.05	0.10 **	0.34 ***	0.42 ***	0.28 ***	0.45 ***	0.44 ***	0.40 ***	--				
8	0.31 ***	0.30 ***	0.25 ***	0.28 ***	0.07	0.08 *	0.34 ***	0.44 ***	0.23 ***	0.50 ***	0.37 ***	0.31 ***	0.68 ***	--			
9	0.14 ***	0.15 ***	0.10 **	0.13 ***	0.06	0.09 *	0.08 *	0.10 **	0.04	0.04	0.10 **	0.05	0.26 ***	0.23 ***	--		
10	0.22 ***	0.20 ***	0.22 ***	0.17 ***	0.05	0.07	0.19 ***	0.22 ***	0.156 ***	0.17 ***	0.16 ***	0.19 ***	0.27 ***	0.26 ***	0.45 ***	--	
11	0.20 ***	0.16 ***	0.18 ***	0.20 ***	0.16 ***	0.18 ***	0.09 *	0.07	0.07	0.04	0.11 **	0.08 *	0.24 ***	0.14 ***	0.36 ***	0.24 ***	--
12	0.15 ***	0.12 ***	0.14 ***	0.15 ***	−0.09	−0.05	0.32 ***	0.33 ***	0.36 ***	0.26 ***	0.28 ***	0.31 ***	0.36 ***	0.34 ***	0.11 **	0.22 ***	0.21 ***
Wave 2																	
1	0.07	0.10 **	0.05	0.03	−0.20 **	−0.17 ***	--										
2	0.13 ***	0.14 ***	0.11 **	0.09 *	−0.13 ***	−0.12 **	0.59 ***	--									
3	0.02	0.05	<0.001	0.01	−0.10	−0.10 **	0.54 ***	0.49 ***	--								
4	0.11 **	0.13 **	0.10 **	0.07	−0.12	−0.12	0.50 ***	0.74 ***	0.56 ***	--							
5	0.11 **	0.18 ***	0.04	0.07	−0.12	−0.13 **	0.41 ***	0.44 ***	0.41 ***	0.45 ***	--						
6	0.06	0.11 **	0.01	0.03	−0.16 ***	−0.17 ***	0.37 ***	0.44 ***	0.37 ***	0.45 ***	0.63 ***	--					
7	0.31 ***	0.32 ***	0.26 ***	0.26 ***	0.05	0.06	0.33 ***	0.43 ***	0.28 ***	0.41 ***	0.47 ***	0.41 ***	--				
8	0.27 ***	0.26 ***	0.25 ***	0.23 ***	0.06	0.05	0.36 ***	0.47 ***	0.30 ***	0.44 ***	0.41 ***	0.31 ***	0.78 ***	--			
9	0.14 ***	0.15 ***	0.11 **	0.14 ***	0.07	0.06	0.23 ***	0.23 ***	0.16 ***	0.15 ***	0.23 ***	0.11 **	0.42 ***	0.35 ***	--		
10	0.16 ***	0.19 ***	0.15 ***	0.10 **	−0.05	−0.02	0.34 ***	0.33 ***	0.23 ***	0.28 ***	0.39 ***	0.26 ***	0.42 ***	0.37 ***	0.56 ***	--	
11	0.16 ***	0.16 ***	0.14 ***	0.14 ***	0.11 **	0.12 **	0.14 ***	0.13 ***	0.15 ***	0.10 ***	0.17 ***	0.07	0.28 ***	0.22 ***	0.44 ***	0.38 ***	--
12	0.09 *	0.09 *	0.09 *	0.06	−0.15 ***	−0.13 ***	0.41 ***	0.38 ***	0.37 ***	0.32 ***	0.35 ***	0.28 ***	0.40 ***	0.37 ***	0.21 ***	0.34 ***	0.21 ***

Notes: * *p* < 0.05; ** *p* < 0.01; *** *p* < 0.001. Abbreviations: Avo = Avoidance; COVID-19 = coronavirus disease 2019; IES-6 = Impact of Event Scale, abbreviated form; Int = Intrusion GAD-7 = General Anxiety Disorder-7; Hyp = Hyperarousal; PHQ-9 = Patient Health Questionnaire-9.

**Table 4 behavsci-11-00076-t004:** Repeated measures ANOVA testing interactions of distress between waves and significant sociodemographics ^a^.

	IES-6	1. Adherence to Hygiene Measures	3. Compliance with Social Distancing
	*F* (df), *η*^2^	*F* (df), *η*^2^	*F* (df), *η*^2^
Waves × Age	0.22 (2, 667), 1.43 × 10^−4^	0.15 (2, 667), 1.11 × 10^−4^	0.66 (2, 666), 5.26 × 10^−4^
Waves × Gender	2.23 (1, 668), 7.06 × 10^−4^	0.02 (1, 668), 7.40 × 10^−6^	3.53 (1, 667), 0.001
Waves × Having children	0.10 (2, 667), 6.71 × 10^−5^	0.56 (2, 667), 4.06 × 10^−4^	1.44 (1, 666), 0.001
Waves × Medical condition	0.78 (1, 666), 3.06 × 10^−4^	1.95 (1, 668), 7.13 × 10^−4^	0.08 (1, 667), 2.39 × 10^−5^

Notes: ^a^ Statistical significance of sociodemographics was established in previous between-subjects effects (see main text). Abbreviations: IES-6 = Impact of Event Scale, abbreviated form.

## Data Availability

The data presented in this study are available on request from the corresponding author. The data are not publicly available due to ethical restrictions.

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
