# Peer review of "Distress, Depression, Anxiety, and Concerns and Behaviors Related to COVID-19 during the First Two Months of the Pandemic: A Longitudinal Study in Adult MEXICANS"

_behavsci, 2021, doi:10.3390/bs11050076_

Round 1
Reviewer 1 Report
In the article “Distress, depression, and anxiety during the first two months of the COVID-19 pandemic: a two-wave prospective study in adult Mexicans” the authors report research on the mental health of Mexicans during the COVID19-pandemic. Data from two waves are presented showing that concerns about the consequences of the pandemic are constantly high across the two measure time points. There is also no substantial change regarding the reported anxiety and depression from April to the end of May 2020. Interestingly, there is even a slight decrease in case of the IES-6 score indicating that there might be a habituation or only an initial reaction that subsided across time. No interactions were found. The authors did a good job in discussing these results and the paper is generally well-written. However, I do have some concern about the presentation of this research that I will outline below:
First, I think the result presentation in the abstract needs to improved. The presentation of the numbers on the distress scores is slightly misleading, since the values decreased across time. I would recommend to focus on the data regarding the concern about the COVID-consequences and the stability or resilience of the sample. Also, in the results section the data should be explained in more detail. On page 6 it reads “a mild difference was observed” – the nature/direction of this difference needs to be explained. Likewise, for the results of table 2.
Second, more information is needed regarding the methods. Can you give an item-example for each scale? Did you calculate the reliability scores of the scales? There were dummy codings regarding the scales, but I guess the table-data is based on the mean values (without considering the cut off values) – is that correct? It would be good, if this would be made clear.
Apart from these concerns, I think such research is essential for our understanding of the consequences of the pandemic and a thorough revised version of this paper could be an important contribution to the field of research.
Reviewer 2 Report
The authors' efforts to evaluate psychological impacts from COVID-19 using a longitudinal approach are appreciated. There are potential contributions from some of the findings of your study. I believe that the study has merit, but do recommend some fairly major revisions. For example, the literature requires greater depth and breadth (including appropriate theoretical and empirical support to frame the study and allow for specific research questions and hypotheses), greater detail in the description of the instruments used in the study, further analysis to reveal potential findings within the dimensions of constructs which were only evaluated in terms of overall mean scores in the current manuscript, clarification of wording in some sections, and a more nuanced approach to reporting the longitudinal (two-wave) findings in the context in which the research was conducted and in comparison to other studies in this field (of which there are several).
Positives:
- Naturally, empirical studies adopting longitudinal analysis of psychological health and behaviors during COVID-19 are very welcome, particularly when the research can be contextualized and interpreted based on local (cultural or policy) factors, for example.
- Overall, the findings are clearly reported and accurately interpreted, with measures of effect size included, and analysis of potential interaction effects from demographic variables also considered.
- The study has some statistically significant findings to share, which can align with previous research and, potentially, offer a contribution to our understanding of the change in the impact of the psychological effects of COVID-19 over time.
Negatives:
- There is a lack of a sufficient review of the literature in terms of key concepts related to the study, including the variables measured and reported in the Results section. Specific constructs, need adequately covered in both the Introduction and the Methods section.
- Based on the lack of depth in the literature review, the research purpose/gap, while evident, was not clearly supported by relevant references to theoretical or empirical literature and, as a result, the research questions should be more clear and include hypotheses (given the fact that previous studies can support the inclusion of the constructs measured).
- The methodology is lacking in clarity in some areas, and the variables are not described in adequate detail. For example, the process of designing, piloting, and validating the authors’ self-designed questionnaire is not explained in this manuscript or their previous publication. Furthermore, details regarding the validity, reliability, and sample items of instruments adopted from other sources are missing in the Methods section. These variables are also not sufficiently covered by the Introduction (literature review)
- The rationale for the analysis of data is not presented clearly in some parts. For example, it is unclear why the dimensions, specific items, or sub-scales of the standardized instruments (IES-6, PHQ-9, and GAD-7) were not evaluated. There is no table for the results of correlation analysis.
- Interpretation of all reported findings should be provided. While there was no interaction effect for age, gender, having children, or pre-existing medical conditions and reported distress between the two waves, the reasons for the lack of differences among these demographic variables is not explained.
- Suggestion: In order to evaluate the effect of demographic variables on the measured outcomes, it might be possible to perform regression analyses in order to evaluate both the potential contribution of these factors, as well as compute an overall R2.
Suggestions for the author, by section.
Title:
- The use of “during” could be misleading. Including “longitudinal” in the title would be more impactful.
- Consider whether or not the study can be classified as “prospective” in terms of the two month duration (relatively short), the lack of pre-COVID findings (baseline measure), and potential biases introduced by the sampling frame.
- Behavioral factors, evaluated by the “Concerns and Behaviors” questionnaire, are not included in the title, which may leave the impression that the article does not clearly fit within the aims and scopes of the Journal.
Abstract:
- The abstract should not include sub-headings, as per Journal policy.
- Reference to the “first two months of the pandemic” is unclear and requires contextualization. For example, COVID-19 was already a pandemic at the time of the first wave.
- The conclusions are not clear in terms of HOW the study adds to research and practice.
Keywords:
- I suggest adding “two-wave design” to the list
- Introduction:
- The demographic variables listed (lines 31-36) are integral to your development of a research gap, selection of research instruments, and interpretation/comparison of the findings. As such, this section should be expanded substantially. Elaboration on prior findings from these cited sources is needed
- Please cite the journal article (lines 40-41), rather than asking the reader to search for it.
- Regarding the findings of the abovementioned article, it is unclear what “dynamics” (line 41) were uncovered.
- The content on lines 42-44 is too similar to the original article and should be paraphrased more carefully.
- The content contained in the first paragraph on page 2 is the type of elaboration which is useful and should be included for other variables of interest, and those which are adopted as instruments in this study.
- There are no theoretical frameworks and unclear empirical findings to support the research questions. The research questions should be more clearly stated and could include specific hypotheses, particularly given the results of previous studies.
- The research gap can be more forcefully argued, with greater support from previous research; namely, the need for longitudinal evaluation of changes in the psychological impact of COVID-19 in the Mexican context.
- Materials and Methods:
- Line 19 is not a full sentence. A more clear overview of the study design should be included here.
- The selection of participants can be more clear (lines 80-86). How can you determine whether or not the sample is representative or free of bias due to the sampling frame. The sampling procedures themselves should also be more clearly stated.
- It is unclear why the demographic information was asked of participants on both occasions, as their age, gender, and occupation are unlikely to have changed (lines 93-94). Furthermore, it should be more clearly stated how the participants’ information was anonymized during the original data collection and the subsequent follow-up (second wave) survey.
- The order of measures (page 3) is consistent with the first research question, but the Results does not report these measures in the same order. The order should be consistent throughout the manuscript.
- It is unclear why the IES-6, PHQ-9, and GAD-7 instruments were not evaluated by subscale or items, while this was done for your self-designed questionnaire.
- Reliability, validity, and sample items should be provided for each dimension of each construct in this section.
- The subheadings for 2.4 through 2.7 should reflect the general construct (i.e., distress, depression, or anxiety) instead of the specific instrument, which can be mentioned in the description.
- The process of developing and validating your self-designed instrument is not detailed enough. There should be information on what theories and empirical research informed the development of the subscales and specific items, how the items were generated, any piloting of the study, the qualifications of the designers, as well as reliability and validity measures. Was CFA or other approaches used to evaluate the latent structures of the measure?
- In terms of the statistical analyses, the use of multiple t-tests is discouraged, unless the significant level can be adjusted to account for potential inflation of the Type I Error Rate. This should not change the significance of the results, given approximations based on the t values, but it should be done to provide a conservative representation of the findings.
- A citation for the interpretation of the effect size statistics (such as Cohen) should be given here, since a η2 of .01 is considered small.
- Results:
- A table of demographic statistics by construct would be useful for the reader.
- While correlation analysis and specific correlations are mentioned, they are not included. I would expect that the five constructs included in the “Concerns and Behaviors” questionnaire and the three standardized psychometric scales would be included in a correlation matrix.
- Please take care in the wording used in the manuscript. Some parts are not clearly or accurately expressed, such as the term “exercised” (line 208).
- Due to the lack of any significant interaction effects, there is no need for Table 3, as the results can simply be reported in the text.
- However, interpretation of these insignificant findings is lacking and should appear in this part of the manuscript.
- Discussion:
- It is unclear how the percentage change in distress was calculated (lines 214-216). At this point the actual effect size should be reported.
- The direction of the changes (positive or negative) for the cited studies (lines 222-225) should be clearly stated.
- Consider an alternative explanation for the low correlation of depressive symptoms with other factors (lines 225-227). Having hypotheses would help in evaluating the results and interpreting them in a more logical fashion. An example of an alternative explanation could be that the measurements adopted for the study did not capture the behaviors or experiences of respondents, or that some biases existed in the sampling process.
- Be careful of the content on lines 233-234 which is very similar to the original source.
- More limitations regarding the length of the study should be considered. Would a shorter or longer duration between waves yield different results?
- Interpretation of the results in terms of precise events occurring between the two waves, in the context of Mexico, would strengthen your contribution.
References:
- References should be checked for proper formatting according to journal policy. For example, article names should be in lower-case, dates should be bold, and volume numbers should be in italics.
Round 2
Reviewer 2 Report
Dear Authors,
Your attention to my several comments is appreciated. I believe that your manuscript has improved. However, there are still a few points that I believe remain to be addressed:
- The variables (DVs) should be defined and linked to prior studies more explicitly and at length in the introduction.
- The RQs and hypotheses are welcome, but they should be: cited with supporting theoretical or empirical evidence, consistently used to organize the article, and mentioned one by one in the Results and Discussion.
- The development process of your "ex profeso" survey requires more detail and evidence of validation.
- Correlations among all DVs should be provided in a matrix.
- Care in interpreting within-subjects (longitudinal) and between-subjects (cross-sectional) effects, as well as moderation effects, should be taken.
- The development of a literature review (with or without theoretical framework) needs to be completed. Even exploratory research requires an clear research goal/rationale, based on a review of the literature in order to select relevant variables. The rationale for the selection of your DVs is based on your knowledge and experience. As such, please provide more details on how you came to these conclusions/decisions.

Author Response
Please, see the attachment.

Round 3
Reviewer 2 Report
I appreciate that you took the time to address some of my prior concerns. I will respond to these points in the attached file and suggest a few areas that could use your further consideration and attention.
In sum, there are some points which could be further improved upon. I believe with a bit more effort, the contribution of this article could be more apparent. The main recommended improvements include:
1) Theoretical/empirical grounding of the study and clear definition and elaboration of the key research variables in the Introduction.
2) Better support for the hypotheses.
3) More clear description of the development of the scale used in the study, with basic reporting of statistics related to internal consistency, for example.
4) More use of signposting (or sub-headings) to indicate which research question is being addressed in the Results and Discussion sections.
5) Consideration of deeper analysis of the data through analysis of the dimensions of the IES-6 (intrusion, avoidance, and hyperarousal).
Please see the attached file for further details.
